# Preparation of Terpenoid-Invasomes with Selective Activity against *S. aureus* and Characterization by Cryo Transmission Electron Microscopy

**DOI:** 10.3390/biomedicines8050105

**Published:** 2020-05-01

**Authors:** Bernhard P. Kaltschmidt, Inga Ennen, Johannes F. W. Greiner, Robin Dietsch, Anant Patel, Barbara Kaltschmidt, Christian Kaltschmidt, Andreas Hütten

**Affiliations:** 1Thin Films & Physics of Nanostructures, Bielefeld University, Universitätsstrasse 25, 33615 Bielefeld, Germany; b.kaltschmidt@uni-bielefeld.de (B.P.K.); ennen@physik.uni-bielefeld.de (I.E.); 2Department of Cell Biology, Bielefeld University, Universitätsstrasse 25, 33615 Bielefeld, Germany; johannes.greiner@uni-bielefeld.de (J.F.W.G.); barbara.kaltschmidt@uni-bielefeld.de (B.K.); c.kaltschmidt@uni-bielefeld.de (C.K.); 3Fermentation and Formulation of Biologicals and Chemicals, Bielefeld University of Applied Sciences, Interaktion 1, 33619 Bielefeld, Germany; robin.dietsch@fh-bielefeld.de (R.D.); anant.patel@fh-bielefeld.de (A.P.); 4Molecular Neurobiology, Bielefeld University, Universitätsstrasse 25, 33615 Bielefeld, Germany

**Keywords:** terpenoids, invasomes, thymol, menthol, camphor, cineol, *S. aureus*, *E. coli*, bactericidal

## Abstract

Terpenoids are natural plant-derived products that are applied to treat a broad range of human diseases, such as airway infections and inflammation. However, pharmaceutical applications of terpenoids against bacterial infection remain challenging due to their poor water solubility. Here, we produce invasomes encapsulating thymol, menthol, camphor and 1,8-cineol, characterize them via cryo transmission electron microscopy and assess their bactericidal properties. While control- and cineol-invasomes are similarly distributed between unilamellar and bilamellar vesicles, a shift towards unilamellar invasomes is observable after encapsulation of thymol, menthol or camphor. Thymol- and camphor-invasomes show a size reduction, whereas menthol-invasomes are enlarged and cineol-invasomes remain unchanged compared to control. While thymol-invasomes lead to the strongest growth inhibition of *S. aureus*, camphor- or cineol-invasomes mediate cell death and *S. aureus* growth is not affected by menthol-invasomes. Flow cytometric analysis validate that invasomes comprising thymol are highly bactericidal to *S. aureus*. Notably, treatment with thymol-invasomes does not affect survival of Gram-negative *E. coli.* In summary, we successfully produce terpenoid-invasomes and demonstrate that particularly thymol-invasomes show a strong selective activity against Gram-positive bacteria. Our findings provide a promising approach to increase the bioavailability of terpenoid-based drugs and may be directly applicable for treating severe bacterial infections such as methicillin-resistant *S. aureus*.

## 1. Introduction

Terpenes are secondary plant metabolides with aromatic characters found in the oil fraction of various plants, where they serve for protection against predators or pathogens [1,2]. Notably, the oxygenated derivatives of terpenes, so-called terpenoids, have strong antioxidant and anti-inflammatory as well as antimicrobial properties [3,4]. For instance, the terpenoid thymol was reported to attenuate allergic airway inflammation in mice [5] and inhibit lipopolysaccharide-stimulated inflammatory responses via down-regulation of the transcription factor NF-κB [6]. Additionally, the terpenoid cineol was shown to inhibit pro-inflammatory signaling mediated by NF-κB [7], while simultaneously potentiating IRF3-mediated antiviral responses [8]. In addition, cineol reduced production of mucus in a human ex vivo model of late rhinosinusitis [9]. Regarding the antimicrobial properties of terpenoids, thymol was reported to have direct bactericidal effects against *S. aureus* and *S. epidermidis* [10]. The terpenoids menthol and camphor also showed anti-bacterial activity against different bacterial species such as streptococci or mycobacteria [11,12]. Cineol was also recently demonstrated to display antibacterial activities against pathogenic bacteria present in chronic rhinosinusitis such as *S. aureus* [13]. Despite these promising anti-inflammatory and anti-bacterial properties, pharmaceutical applications of terpenoids against bacterial infection remain challenging due to their poor water solubility and high volatility.

Here, we address this challenge by utilizing liposomal packaging for drug delivery of terpenoids. Liposomes are spherical vesicles of phospholipid bilayers, which are commonly used for encapsulation of drugs and particularly for increasing or allowing their anti-microbial activity [14]. For instance, Aravevalo and colleagues showed an increase in antibiotic activity of ß-Lactam against resistant *S. aureus* after encapsulation within coated-nanoliposomes [15]. In addition, Moyá and coworkers recently demonstrated a bactericidal activity of Cefepime encapsulated into cationic liposomes against *E. coli* [16]. Engel and colleagues assessed the antimicrobial activity of thymol and carvacrol encapsulated into liposomes by thin-film hydration and observed an inhibition of *S. aureus* and *S. enterica* growing on stainless steel [17]. As recently reported by Usach and coworkers, pompia essential oil (containing limonene and citral) as well as citral itself were successfully loaded into liposomes via hydration followed by ultrasonic disintegration. The respective encapsulated terpenoids had antimicrobial properties against different bacterial species such as *S. aureus* or *P. aeruginosa* [18]. A nanoemulsion comprising eucalyptus oil obtained by ultrasonic emulsification was also shown to have antibacterial activity against *S. aureus* [19]. Cui and colleagues further observed an antibacterial effect of cinnamon oil (with eugenol being its main compound) encapsulated into liposomes against methicillin-resistant *S. aureus* (MRSA) alone or cultivated as a biofilm [20]. In addition to increasing the anti-microbial activity of drugs, encapsulation into liposomal structures was already described to enhance the anti-inflammatory capacity of terpenoids. In this regard, nanostructured lipid carriers encapsulating thymol were shown to provide a sustained release of the terpenoid as well as an increase in anti-inflammatory activity within mouse models of skin inflammation [21]. In addition to nanostructured lipid carriers, terpenoids can also be delivered by encapsulation into polymeric nanostructured systems or by molecular complexation [2]. In addition to increasing stability of encapsulated compounds, terpenoid-encapsulation systems are widely accepted to be non-cytotoxic and enhance the antioxidant and anti-inflammatory activities of terpenoids ([18,21,22,23] reviewed in [2] and [24]). For instance, Manconi and colleagues reported the liposomal formulation of thymus essential oil to be highly biocompatible and to counteract oxidative stress in keratinocytes [22]. Thymol encapsulated in nanostructured lipid carriers further showed anti-inflammatory activity in different mouse models of skin inflammation in vivo [21].

In the present study, we took advantage of a liposome-based system, the so-called invasomes, for encapsulating the terpenoids thymol, menthol, camphor and cineol (Figure 1). Invasomes are liposomes composed of unsaturated phospholipids, small amounts of ethanol, terpenes and water [25]. In our present approach, terpenoid-invasomes were produced via extrusion of a solution comprising the respective terpenoid solved in ethanol, while soybean lecithin served as a lipid source. We aimed to characterize the produced invasomes in terms of their lamellarity, size and bilayer thickness using cryo transmission electron microscopy (Cryo TEM). Although being commonly utilized for the formulation of invasomes [26,27], terpenoids mostly serve as permeation enhancer for transdermal delivery and bioavailability rather than being bioactive components [2]. In the present study, we focused on the production of invasomes with terpenoids as the bioactive components to inhibit bacterial cell growth. With respect to pharmaceutical applications for treating bacterial infection, we therefore assessed potential bactericidal effects of our produced terpenoid-invasomes against *S. aureus* and *E. coli*.

## 2. Materials and Methods

### 2.1. Preparation of Invasomes and Encapsulation of Terpeonoids

For production of invasomes, 20 mg of the respective terpenoid thymol, menthol, camphor or cineol (Caesar and Loretz GmbH, Hilden, Germany) was solved in 1.32 mL of 99.6% Ethanol. This mixture was subsequently added to 40 mL of 0.9% saline to reach a final concentration of 0.5 mg terpenoid per mL. A quantity of 100 mg of soybean lecithin (Lipoid S 100, Batch 579000-1160713-01/704, Lipoid GmbH, Ludwigshafen am Rhein, Germany) was added to 2 mL of this solution comprising the terpenoid, ethanol and saline followed by vortexing for 5 min to reach a homogenous solution. For control formulation, a solution without terpenoids was applied. Invasomes were formed by extrusion using the Avanti Mini-Extruder (Avanti Polar Lipids, Alabaster, AL, USA) by passing 1 mL of final solution 10 times back and forth through a polycarbonate membrane with 100 nm pores.

### 2.2. Cryo Transmission Electron Microscopy (Cryo TEM)

For Cryo TEM analysis, 3 µL of the respective invasome-dispersion produced as described above was placed on a TEM copper grid (Quantifoil Micro Tools GmBH, Großlöbichau, Germany). Plunging into liquid ethane using Leica EM GP (Leica Microsystems, Wetzlar, Germany) with 80% moisture, 10 s pre-blotting time, 3 s blotting time and 20 °C temperature was followed by transporting the samples to the cryo transfer station (Fischione Intruments, Export, PA, USA) in liquid nitrogen. Analysis was done at the OWL Analytic Center using Jeol JEM 2200 FS (JEOL Ltd., Tokyo, Japan) operated at 200 kV.

### 2.3. Determination of Zeta Potentials

Zeta potentials were measured using Beckmann Coulter Delsa Nano C Particle Analyzer (Beckman Coulter, Brea, CA, USA) in a flow cell after dilution of samples with water from 50 mg/mL to 500 µg/mL. Measurements were repeated ten times.

### 2.4. Evaluation of Invasome Size and Bilayer Thickness

Area and bilayer thickness of produced invasomes were measured using FIJI [28] by utilizing Cryo TEM images. Briefly, the area of every invasome was marked with the circular selection tool and the measurement function was applied to calculate the area of the selections followed by calculation of the radius. For assessing bilayers thickness, the segmented area selection tool of FIJI was used followed by the straighten function of FIJI to obtain straight selection and calculation of a line profile. Respective line profiles showed a clear dent at the bilayer position and thickness was measured at half maximum. A Gauss distribution was fitted to all histograms.

### 2.5. Growth-Inhibition Zone Assay

Overnight suspension cultures of *S. aureus* (*Staphylococcus aureus* Rosenbach 1884, DSM 24167, German Collection of Microorganisms and Cell Cultures GmbH (DSMZ), Braunschweig, Germany) were inoculated on Brain Heart Infusion Broth (BHI) agar plates (Sigma-Aldrich Corporation, Merck KGaA, Darmstadt, Germany). Filter plates loaded with 10 µL of respective invasome-dispersion were placed on the BHI agar plates followed by incubation overnight at 37 °C. Diameters of the growth-inhibition zones were measured and calculated using FIJI [28].

### 2.6. Analysis of Encapsulation Efficiency and Loading Capacity

Encapsulation efficiency (EE %) was analyzed as described in [29]. EE% was calculated by (total drug added − free non-entrapped drug) divided by the total drug added. Loading capacity (LC) was calculated as the amount of drug loaded per unit weight of total invasomes (weight of lipids). Free drug was separated from invasome preparation by ultrafiltration (1000× *g* for 30 min at 4 °C) using an Amicon Centricon device with a molecular weight cut-off of 30,000. Drug stocks for measurement were prepared in methanol for HPLC (VWR) at a concentration of 1 mg/mL. A linear dilution was prepared in methanol and absorbance was measured at 276 nm using an Ultrasspec 2100 pro photometer (Amersham Biosciences, Little Chalfont, UK). Linearity was proven between 0.0015 mg/mL and 0.025 mg/mL thymol. EE% for thymol was calculated as (0.5 mg/mL−0.0041 mg/mL)/0.5 mg/mL = 0.9918 resulting in 99.18% EE. LC% was calculated as (0.5 mg/mL−0.0041 mg/mL)/(50 + 0.5) mg/mL= 0.01, which equals 1%.

### 2.7. Flow Cytometric Measurement of Cell Death

For flow cytometric measurement of cell death, *S. aureus* or *E. coli* (*Escherichia coli* DH5-Alpha) were exposed to respective invasome-dispersions (0.5 mg/mL, 1 mg/mL, 2 mg/mL final terpenoid concentration) for 24 h. Cells were fixed with 0.4% PFA for 20 min followed by staining with 1 µL/mL propidium iodide (PI, Sigma Aldrich) for 10 min. PI-stained bacterial cultures were analyzed using a Beckman Coulter Gallios Flow Cytometer (Beckman Coulter) followed by data analysis with Kaluza software (Beckman Coulter).

### 2.8. Statistical Analysis

For assessment of lamellarity and size of invasomes encapsulating a distinct terpenoid, up to 200 invasomes per Cryo TEM image were measured in 3–4 representative Cryo TEM images. For evaluation of bilayer thickness, up to 30 invasomes were measured in 3–4 representative Cryo TEM images for each of the different terpenoid-invasomes. Examples of small representative sections of original cryo electron micrographs used for measuring invasomes size are included within the respective figures, details of measuring are described above. Statistical analysis was performed using Graph Pad Prism (GraphPad Software, San Diego, CA, USA). The *p* value is a probability, with a value ranging from zero to one. The first step is to state the null hypothesis, here that the terpenoids do not affect the size of the invasomes and all differences in size are due to random sampling. The *p*-value is the probability of obtaining results as extreme as the observed results of a statistical hypothesis test, assuming that the null hypothesis is correct. The *p*-value is used as an alternative to rejection points to provide the smallest level of significance at which the null hypothesis would be rejected. A smaller *p*-value means that there is stronger evidence in favour of the alternative hypothesis. For analysis of lamellarity, * *p* < 0.05 was considered significant (Mann–Whitney test, one-tailed). For analysis of invasome size, *p* < 0.0001 was considered significant (unpaired t-test, one-tailed). Growth-inhibition zone assay was performed as biological triplicate and Graph Pad Prism served for statistical analysis with * *p* < 0.05 being considered significant (Mann–Whitney test, one-tailed).

## 3. Results and Discussion

### 3.1. Succsessfull Preparation of Invasomes Encapsulating the Terpenoids Thymol, Menthol, Camphor and Cineol

To encapsulate terpenoids into invasomes, we aimed to produce liposomes by extrusion of a homogenous solution comprising the respective terpene solved in ethanol, 0.9% saline and soybean lecithin. Extrusion of a solution without terpenoids served as control and resulted in the formation of invasomes, as visualized by cryo transmission electron microscopy (Cryo TEM) (Figure 2A).

We next applied the terpenoids thymol, menthol, camphor or cineol for production of invasomes (Figure 2B). Cryo TEM micrographs showed the presence of invasomes comprising thymol (Figure 2C), menthol (Figure 2D), camphor (Figure 2E) or cineol (Figure 2F) after extrusion. Interestingly, Cryo TEM allowed us to observe multilamellar membrane boundaries in all five approaches (Figure 2A,C–F).

During characterization of the newly produced invasomes via Cryo TEM, we observed that encapsulation of thymol and camphor resulted in a significant shift towards unilamellar vesicles. We suggest that terpenoids such as thymol might decrease membrane fluidity and thus lead to more unilamellar liposomes. On the contrary, cineol-invasomes revealed a similar distribution between unilamellar and bilamellar vesicles comparable to control. Furthermore, cineol-invasomes showed a similar size to control-invasomes, while encapsulation of thymol and camphor led to significantly smaller invasomes and menthol-comprising invasomes were significantly enlarged compared control. We suggest this observation to be based on the elevated water solubility of cineol (3500 mg/L) compared to camphor (1600 mg/L), thymol (900 mg/L) and menthol (420 mg/L).

Analysis of Zeta potentials revealed approximately neutral potentials of control invasomes (−2 ± 5 mV, Figure 3A). We also observed approximately neutral Zeta potentials for invasomes encapsulating the terpenoids thymol (−3 ± 6 mV, Figure 3B), menthol (−1 ± 5 mV, Figure 3C), camphor (−2 ± 5 mV, Figure 3D) or cineol (0 ± 5 mV, Figure 3E).

In terms of invasomal stability, high Zeta potentials of at least ± 20 mV are generally considered as an indicator for electrostatical and steric stabilization of invasomes [30,31]. Although all produced terpenoid-invasomes showed neutral Zeta potentials in the present study, we observed the presence of stable invasomes using Cryo TEM after extrusion. A limitation of our study was that the low values of zeta potential could only be measured with low precision, e.g., 3 mV ± 6 for thymol-containing invasomes. Furthermore, measurements of Zeta potentials in nano carrier systems such as invasomes are hampered by measuring limitations arising in diluted samples. Hence, plenty of parameters which influence zeta potentials such as viscosity, pH, and dielectric constant are not correctly reflected in diluted samples [32]. Cryo TEM, to the best of our knowledge, does not have these limitations, since samples with much higher concentrations of invasomes could be analysed in their native diluent.

In accordance with our findings, Sebaaly and colleagues reported neutral Zeta potentials of −3.9 ± 1.9 mV for eugenol-loaded Lipoid S100-liposomes prepared by ethanol injection method. Although the authors demonstrated an increase in liposome size and size distribution after storage in aqueous suspension at 4 °C for 2 months, encapsulation efficiencies of eugenol (86.6%) were unchanged [33]. We suggest that encapsulation efficiencies of our produced invasomes may not be affected over time, despite their neutral Zeta potentials.

In addition to providing an increased bioavailability and a more controlled drug release, our approach may also facilitate topical administration of thymol-invasomes due to the high permeability rate of invasomes through the skin [2,21,24].

Notably, extrusion of solution without addition of ethanol for solving the terpenoid of interest did not result in the formation of invasomes (data not shown). In summary, we successfully prepared invasomes encapsulating the terpenoids thymol, menthol, camphor and cineol.

### 3.2. Encapsulation of Terpenoids Significantly Changes Lamellarity and Size of Invasomes without Affecting Bilayer Thickness

We next characterized the produced invasomes in more detail in terms of their lamellarity, size and bilayer thickness. For investigation of lamellarity, we determined individual types of lamellar phase lipid bilayers ranging from one lipid bilayer (MLV1) up to eight lamellar phase lipid bilayers (MLV8) (Figure 4A). Cryo electron micrographs (see also Figure 2) served for determination of the individual types of lamellar phase lipid bilayers, which we present in relation to their distribution. Notably, we observed strong differences in lamellarity of invasomes depending on the added terpenoids and in comparison to control. Without the addition of a terpenoid (control, Figure 4B), mostly unilamellar (37 ± 8%) and bilamellar vesicles (37 ± 13%) were formed (* *p* < 0.05, Figure 4B) and showed almost equal proportions (*p* = 0.4 was not considered significant, Mann–Whitney test, one-tailed).

On the contrary, a shift towards 63 ± 11% unilamellar vesicles (MLV1, Figure 4C) and a significantly decreased amount of 26 ± 7% bilamellar vesicles (MLV2, * *p* < 0.05, Figure 4C) were observed for invasomes comprising of thymol. Production of invasomes with menthol resulted in a significantly increased amount of unilamellar vesicles (54 ± 18%) compared to MLV3–8 (* *p* < 0.05), but no significant changes between the amounts of MLV1 and MLV2 (*p* = 0.0571 was not considered significant, Mann–Whitney test, one-tailed, Figure 4D). Similarly to thymol-invasomes, encapsulation of camphor resulted in mostly unilamellar vesicles (59 ± 10%, * *p* < 0.05) and significantly decreased amounts of bilamellar vesicles (32 ± 12%, * *p* < 0.05, Figure 4E). Interestingly, invasomes containing cineol revealed a similar lamellarity as the control with a similar distribution between unilamellar (43 ± 3%) and bilamellar vesicles (45 ± 2%, *p* = 0.2000 was not considered significant, Mann–Whitney test, one-tailed, Figure 4F). In addition, no MLV6–8 were observable in invasomes encapsulating cineol (Figure 4F).

In addition to their lamellarity, we measured and calculated the size of the produced invasomes (Figure 5A). All invasomes including the control showed a large distribution in size but also specific changes according the encapsulated terpenoid. Compared to control invasomes showing a size distribution from about 20 up to 80 nm radius and a mean of 40 ± 15 nm (Figure 5B), thymol-containing invasomes revealed a significantly smaller radius of 33 *±* 18 nm (Figure 5C, *** *p* < 0.0001 was considered significant, unpaired t-test, one-tailed). Preparation of invasomes with camphor also resulted in a significantly smaller invasomes size (30 *±* 16 nm radius, Figure 5E) compared to control (*** *p* < 0.0001, unpaired t-test, one-tailed). On the contrary, menthol-comprising invasomes revealed a significantly increased radius of 58 *±* 22 nm (Figure 5D) compared to all other terpenoid-encapsulating invasomes and control invasomes (*** *p* < 0.0001, unpaired t-test, one-tailed). Interestingly, although their distribution showed a peak at 35–40 nm radius, the size of invasomes with cineol (43 *±* 17 nm radius, Figure 5F) was similar to control (40 ± 15 nm, *p* = 0.13 was not considered significant, unpaired t-test, one-tailed).

As a third parameter for characterization of our newly produced invasomes, thickness of the liposomal bilayer was measured by evaluating cryo-electron micrographs (Figure 6A). We observed no significant differences (Mann–Whitney test, one-tailed) in the liposomal bilayer thickness of control invasomes (4 ± 0.5 nm, Figure 6B) compared to menthol-invasomes (4 *±* 0.3 nm, *p* = 0.2482, Figure 6D) and invasomes encapsulating camphor (4 *±* 0.5 nm, *p* = 0.2987, Figure 6E). Invasomes produced with thymol revealed a slightly but significantly increased bilayer thickness of 5 *±* 1 nm (Figure 6C) compared to control (** *p* < 0.01), camphor-invasomes (* *p* < 0.05) and cineol-invasomes (** *p* < 0.01, Mann–Whitney test, one-tailed). In contrast, cineol-comprising invasomes showed a minor decrease in bilayer thickness (4 *±* 0.5 nm, Figure 6F) in comparison to the other approaches.

Taken together, invasomes with terpenoids showed a shift towards unilamellar vesicles, except for cineol with a similar distribution between unilamellar and bilamellar vesicles comparable to control. While the bilayer thickness of invasomes was comparable in all approaches, preparation of invasomes with thymol and camphor led to significantly smaller invasomes compared control. On the contrary, menthol-comprising invasomes were significantly enlarged and we observed the radius of cineol-invasomes to be comparable to control.

### 3.3. Invasomes Encapsulating Thymol, Camphor and Cineol Show Bactericidal Activity against S. aureus in a Growth-Inhibition Zone Assay

After successfully producing and characterizing terpenoid-comprising invasomes, we assessed their potential bactericidal activity against *S. aureus* in a growth-inhibition zone assay. After exposure of *S. aureus* to different terpenoid-invasomes overnight, we determined the size of the inhibitory zones. Control invasomes without terpenoids did not result in growth inhibition of *S. aureus* (Figure 7A). Compared to unaffected control, we observed a clearly visible growth inhibition of *S. aureus* exposed to invasomes comprising thymol, camphor or cineol (Figure 7B,D,E). Interestingly, invasomes encapsulating menthol did not affect growth of *S. aureus* (Figure 7C). Statistical evaluation of the measured areas of inhibition validated a strong and significant inhibition of bacterial growth by thymol-containing invasomes compared to all other terpenoid-invasomes and control (Figure 7F). However, invasomes comprising camphor or cineol still caused a significant increase in the zone of inhibition compared to control and menthol-invasomes, which revealed no zone of inhibition (Figure 7F).

In Table 1, the particle size of different terpenoid formulations is depicted as measured in Figure 5. When the formulations are sorted from the highest antibacterial activity (thymol) to the lowest (cineol), as measured in Figure 7, it becomes evident that the terpenoids with the highest antibacterial activity have the highest polydispersity index also (Table 1).

### 3.4. Flow Cytometric Analysis of Cell Death Validate High Bactericidal Activity of Thymol-Loaded Invasomes against Gram-Positive S. aureus

To validate the strong bactericidal activity of thymol-invasomes against *S. aureus* in the growth-inhibition zone assay, we also assessed the anti-bacterial activity of thymol-invasomes quantitatively using flow cytometry. Bacterial cell death was measured by the DNA-intercalating dye Propidum Iodide (PI), which is incorporated only by dead cells. Prior to this analysis, we determined the encapsulation efficiency and loading capacity of thymol-invasomes to ensure proper encapsulation of the terpenoid. Here, we observed an encapsulation efficiency of 47 ± 13% as well as a loading capacity of 0.5 ± 0.1% for invasomes encapsulating thymol. Compared to untreated negative control (4–9% cell death), 0.5 mg/mL thymol packaged in invasomes resulted in a profound cell death of 70% (Figure 8A). Exposure of 1 mg/mL invasome-encapsulated thymol even resulted in 75% bacterial cell death (Figure 8B). Since 2 mg/mL thymol packaged in invasomes resulted in only 9% PI-stained *S. aureus* (Figure 8C), we additionally assessed the cell count per second during the flow cytometric measurement. Here, only around 1000 cells/second were observed in the *S. aureus* population treated with 2 mg/mL thymol-comprising invasomes, whereas the cell count for control conditions ranged around 40,000 cells/second (Figure 8D). Treatment of *S. aureus* with 0.5 mg/mL or 1 mg/mL invasome-encapsulated thymol resulted in cell flow of around 33,000 cells/second (Figure 8D). Thus, we suggest that 2 mg/mL thymol packaged in invasomes already resulted in a nearly complete cell death of *S. aureus* prior to PI-staining and following flow cytometric measurements. We conclude invasomes encapsulating thymol to be strongly bactericidal against Gram-positive bacteria such as *S. aureus*.

### 3.5. Thymol-Loaded Invasomes Do Not Affect Survival of Gram-Negative E. coli

Next to Gram-positive bacteria such as *S. aureus,* we assessed the potential anti-bacterial activity of thymol-invasomes against Gram-negative species such as *E. coli*. Notably, 0.5 mg/mL thymol packaged in invasomes resulted in only 0.2% PI-stained dead cells (Figure 9A). Treatment of *E. coli* with 1 mg/mL or 2 mg/mL invasome-encapsulated thymol only led to 6% or 5% cell death (Figure 9B,C). With regards to the low amount of PI-stained cells, we also assessed the cell count per second during the flow cytometric measurement. Here, we observed no relevant effects of the different concentrations of invasome-encapsulated thymol on the growth of *E. coli (*cell count per second, Figure 9D). In summary, the invasomes comprising thymol produced in this study are highly bactericidal to Gram-positive *S. aureus,* but do not affect survival of Gram-negative *E. coli*.

### 3.6. Growth-Inhibition Zone Assay Shows Strong Bactericidal Activity of Cineol Invasomes against E. coli

In addition to thymol-invasomes, we assessed the potential bactericidal activity of invasomes encapsulating menthol, camphor and cineol against *E. coli.* In line with our flow cytometric analysis of cell death, thymol-invasomes revealed no elevated growth inhibition of *E. coli*, similarly to control-invasomes without encapsulated terpenoids (Figure 10A,B). While menthol-invasomes led to a slight growth inhibition of *E. coli* (Figure 10C)*,* invasomes encapsulating camphor showed no bactericidal activity against *E. coli* (Figure 10D). Notably, exposure of *E. coli* to invasomes loaded with cineol resulted in a strong and significant growth inhibition (Figure 10E,F).

We next determined the potential bactericidal activity of non-extruded terpenoids as a control to our encapsulation approach. In contrast to terpenoids encapsulated in invasomes (Figure 7, Figure 8, Figure 9 and Figure 10), application of non-extruded terpenoids did not result in growth inhibition of *E. coli* or *S. aureus* (data not shown). In summary, the invasomes comprising thymol produced here are highly bactericidal to Gram-positive *S. aureus,* while cineol-invasomes affect the survival of Gram-negative *E. coli* (Figure 7, Figure 8, Figure 9 and Figure 10).

Potential antibacterial mechanisms of invasome formulations with terpenoids are depicted in Figure 11.

Although terpenoids such as limonene, cineole or beta-citronellene have been widely used for formulation of invasomes [26,27], they were mostly applied as permeation enhancer for transdermal delivery and bioavailability and not as bioactive components [2]. In the present study, we focused on the production of invasomes with terpenoids as the bioactive components to inhibit bacterial cell growth. We found our terpenoids-invasomes to be bactericidal against Gram-positive *S. aureus*, with increasing efficiency from cineol- and camphor-invasomes (moderate bactericidal activity) to thymol-invasomes showing the strongest bactericidal effects. These findings are in line with the commonly reported bactericidal activity of thymol, camphor and cineol [10,12,13]. Interestingly, Mulyaningsih and colleagues reported that exposure of MRSA even to high concentrations of cineol does not inhibit multi-resistant *S. aureus*. However, a combination of the terpene aromadendrene with cineol resulted in reduced bacterial cell growth [34]. Extending these findings, we show that encapsulation of cineol into invasomes alone is sufficient for inhibiting growth of *S. aureus* without the application of additional terpenes. In accordance to the strong bactericidal effects of thymol-invasomes observed here, encapsulation of thymol into other nanocarriers such as ethylcellulose/methylcellulose nanospheres was also reported to preserve its anti-bacterial activity against *S. aureus* [35]. In contrast to invasomes encapsulating thymol, camphor and cineol, we observed no anti-bacterial effects for menthol-invasomes against *S. aureus*, which is contrary to the already described bactericidal activity of menthol [11,12]. With regard to the very low water-solubility of menthol, we suggest the invasomal packaging of menthol to be challenging, in turn, affecting its bactericidal activity. In this line, we observed an increased average size of menthol-invasomes compared to all other terpenoid-invasome preparations and control-invasomes lacking terpenoids. In addition polydispersity index as a measurement of the uniformity of invasome size distribution, with a higher value resulting in a broader distribution, was highest with thymol (0.3) and camphor (0.3), suggesting a correlation to antibacterial activity.

Next to Gram-positive bacterial species such as *S. aureus*, we also investigated potential anti-bacterial properties of our invasomes and particularly, thymol-invasomes on Gram-negative *E. coli*. Here, cineol-comprising invasomes led to a strong inhibition of *E. coli* growth, which is in line with our previous observations [13]. In contrast to the strong bactericidal effects against *S. aureus*, cell growth of Gram-negative *E. coli* was not affected by thymol-invasomes. These observations are contrary to the findings by Salvia-Trujillo and colleagues reporting a bactericidal activity of essential oils of thyme (containing thymol) after incorporation into nano-emulsions [35]. In particular, nano-emulsions comprising essential oils of thyme with heterogeneous droplets sizes between 10 nm to 500 nm were shown to reduce growth of *E. coli* [36]. However, the authors applied unfractionated essential oils, suggesting a synergistic action of many terpenes to be necessary for bactericidal activity against Gram-negative species. Interestingly, Trombetta and coworkers demonstrated that *S. aureus* appears to be far more sensitive to thymol than *E. coli* [37], which is in line with our present data. The authors reported a minimal inhibiting concentration of 5.00 mg/mL thymol for *E. coli* [37], suggesting the concentration of up to 2 mg/mL thymol in the invasomes applied here to be not sufficient for inhibition of *E. coli* growth. Furthermore, *S. aureus* is known to secrete pore-forming toxins (PFTs), which were shown to mediate the release of encapsulated clove oil from liposomes. In particular, Cui and coworkers reported that PFTs form pores within the liposome membranes, allowing release of the encapsulated clove oil and facilitating its antibacterial activity. On the contrary, liposomal packaged clove oil had no bactericidal effects on *E. coli*, which does not secrete PFTs and thus prevents the release of antibacterial essential oil from the invasome [38]. Our present observations may suggest a similar mechanism for thymol-invasomes leading to its selective activity against *S. aureus* (Figure 11). In addition, electrophoretic mobility measurement revealed a harder surface of *E. coli* compared to *S. aureus* [39]. The softer surface of *S. aureus* mainly comprising peptidoglycan may facilitate entry of thymol-invasomes into the bacterial cells more easily compared to Gram-negative *E. coli* (Figure 11). Accordingly, we achieved a highly efficient killing of *S. aureus* with only 0.5 mg/mL thymol encapsulated in invasomes in the present study (Figure 11).

In summary, we demonstrate the successful production of invasomes encapsulating thymol, menthol, camphor or cineol and show a strong selective activity of thymol-invasomes against Gram-positive *S. aureus*. As a further benefit of our approach, encapsulation of terpenoids into nanocarrier systems such as invasomes is suggested to increase stability and protect against environmental factors causing degradation [2,33,40]. Here, liposomes composed of lipoid S100 and cholesterol were reported to retain considerable concentrations of isoeugenol, pulegone, terpineol, and thymol liposomes even after 10 months [29]. The application of soy lecithin liposomes comprising cinnamon oil was further shown to improve stability of the essential oil and extend the bactericidal action time [20]. In addition, the application of invasomes was particularly found to elevate the stability of the encapsulated compounds (reviewed in [24]).

There are several nanocarrier systems, encapsulating terpenoids, which are systematically reviewed in [2]. The formulation systems encapsulating terpenoids include polymer-based systems such as nano-capsules, nano-particles, nano-fibers and nano-gels. Furthermore, lipid-based systems are frequently used (67% of the formulations), presumably due to the low toxicity. A subgroup of lipid systems are the vesicular systems, which include invasomes. The most investigated biological activity of terpenoids in nano carrier systems is the anti-inflammatory action. Invasomes were used as anti-acne treatments, hypertension treatment and photosensation therapy [2].

Antimicrobial activity was reported with nano capsules with essential oils from lemon grass, nano emulsions with tea tree oil and penetration-enhancing vehicles with essential oil from Santolina insularis. Here, we report a novel antimicrobial application of invasome formulations with terpenoids.

We conclude that our findings might provide a promising approach to increase the bioavailability of terpenoid-based drugs and might be applicable for treating severe bacterial infections such as MRSA in the future. In this regard, the major treatment aims of our formulations include a broad spectrum of applications, ranging from mucosal infections in airway diseases to systemic infections such as sepsis. In this direction, we have previously shown that patients with chronic rhinosinusitis have increased levels of S. aureus-containing biofilms in the nose [13]. Growth of *S. aureus* biofilms on the nasal mucosa could be inhibited by 1,8-cineol. Here, we extend these findings to thymol-containing invasomes, which have superior antibacterial activity than formulations with 1,8-cineol (see Figure 7). Taken together, an invasome formulation as described here, containing thymol might be useful as an aerosol spray for pre-operative nose cleaning and might have fewer side effects in comparison to disinfectants directly applied on the mucosa. As a general use, it might be envisaged that invasomes containing thymol or other terpenoids could be employed to treat infected surfaces as in nose, lung and skin wounds. Finally, invasomes containing terpenoids might be used in addition or as an alternative to antibiotics.

## Figures and Tables

**Figure 1 biomedicines-08-00105-f001:**
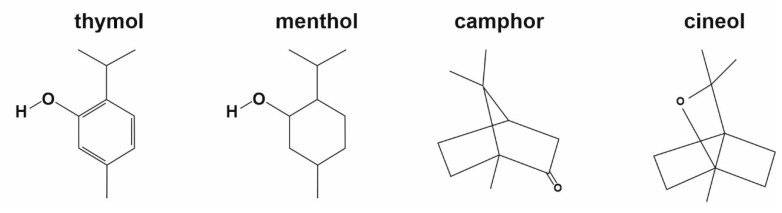
Structural formulas of the terpenoids thymol, menthol, camphor and cineol used for encapsulation into invasomes.

**Figure 2 biomedicines-08-00105-f002:**
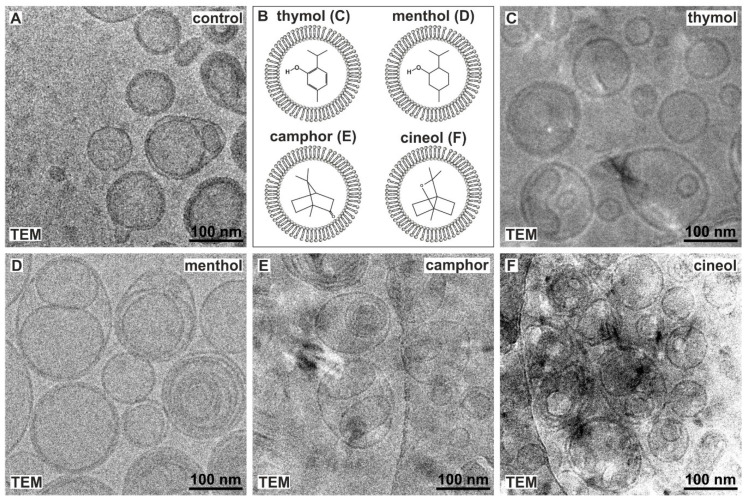
Successful production of invasomes encapsulating thymol, menthol, camphor and cineol by extrusion. (**A**) Cryo transmission electron microscopy (TEM) showing control invasomes without terpenoids (small representative section of original micrograph). (**B**) Schematic view of terpenoids encapsulated by invasomes. Localization of terpenoids within the aqueous phase of the invasome was chosen only for visualization reasons and does not represent their natural localization. (**C**–**F**) Small representative sections of original cryo electron micrographs revealed invasomes comprising thymol (**C**), menthol (**D**), camphor (**E**) or cineol (**F**) after extrusion. TEM: Cryo transmission electron microscopy.

**Figure 3 biomedicines-08-00105-f003:**
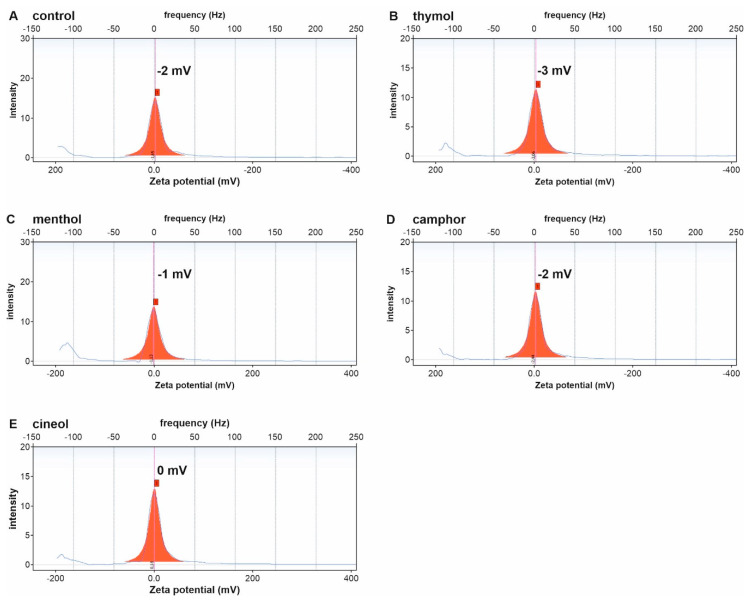
Invasomes without tepenoids (**A**) and encapsulating the terpenoids thymol (**B**) menthol (**C**) camphor (**D**) or cineol (**E**) reveal neutral Zeta potentials.

**Figure 4 biomedicines-08-00105-f004:**
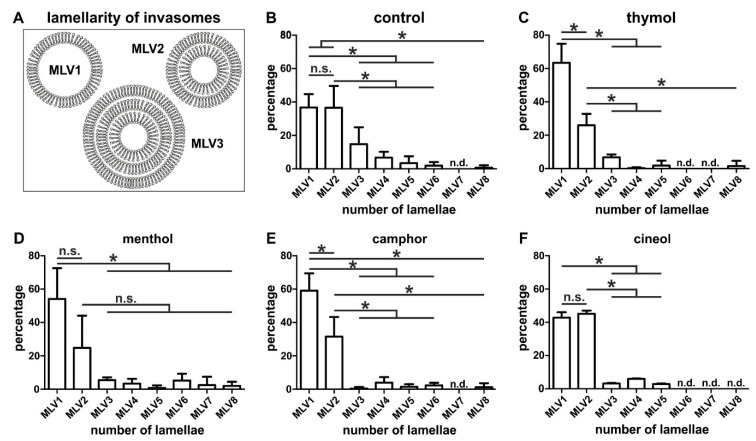
Characterization of lamellarity of the produced terpenoid-comprising invasomes. (**A**) Schematic view of individual types of lamellar phase lipid bilayers. (**B**) Without the addition of a terpenoid (control, **B**), mostly unilamellar and bilamellar vesicles were formed in equal proportions. (**C**–**E**) Invasomes comprising of thymol or camphor showed mostly unilamellar vesicles and a significantly smaller amount of bilamellar vesicles, while menthol-invasomes revealed no changes between MLV1 and MVL2. (**D**) Cineol-invasomes revealed a similar distribution between unilamellar and bilamellar vesicles. Distribution of the individual vesicle types is depicted in relation to their lamellarity measured from respective cryo electron micrographs. * *p* < 0.05 was considered significant (Mann–Whitney test, one-tailed). MLV: Multilamellar vesicles. (n.s. means not significant)

**Figure 5 biomedicines-08-00105-f005:**
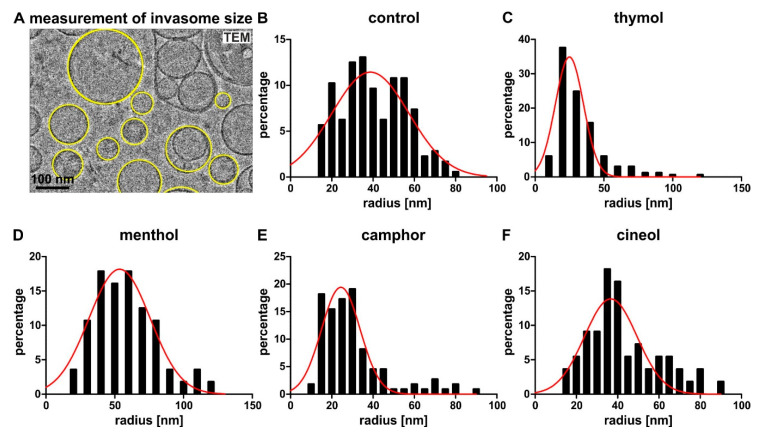
Encapsulation of terpenoids directly affects invasomes size. (**A**) Example of a small representative section of an original cryo-electron micrograph used for measuring invasomes’ size. (**B**) Control invasomes without terpenoid showing a mean radius of 40 ± 15 nm. (**C**) Thymol-comprising invasomes revealed a smaller radius of 33 *±* 18 nm compared to control. (**D**) Production of invasomes with menthol resulted in an increased invasome radius of 59 *±* 22 nm. (**E**) Like thymol, camphor-invasomes also shower a smaller invasome size (30 *±* 16 nm radius) compared to control. (**F**) With a mean radius of 43 *±* 17 nm, the size of invasomes with cineol was similar to control. Frequency plots of the radius distribution of the invasomes. A fit with the Gaussian function is displayed as a red line.

**Figure 6 biomedicines-08-00105-f006:**
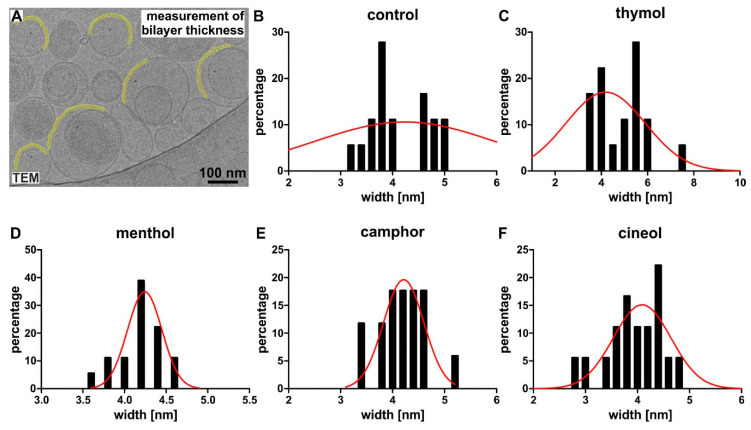
Encapsulation of terpenoids does not affect the bilayer thickness of invasomes. (**A**) Example of a small representative section of an original cryo-electron micrograph used for measurement of the bilayer thickness. (**B**–**F**) A similar liposomal bilayer thickness was observable for control invasomes (4 ± 0.5 nm), thymol-invasomes (5 *±* 1 nm) menthol-invasomes (4 *±* 0.3 nm), camphor-invasomes (4 *±* 0.5 nm) or cineol-invasomes (4 *±* 0.5 nm). Frequency plots of the bilayer thickness distribution of invasomes. A fit with the Gaussian function is displayed as a red line.

**Figure 7 biomedicines-08-00105-f007:**
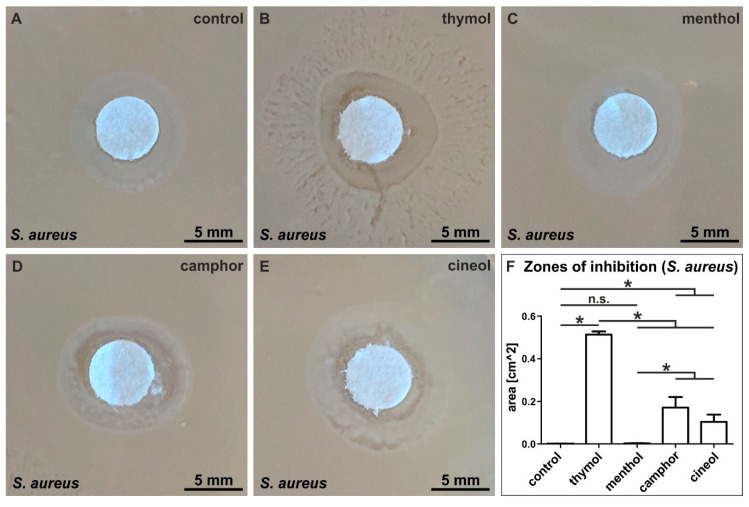
Invasomes encapsulating thymol, camphor and cineol show bactericidal activity against *S. aureus* in a growth-inhibition zone assay. (**A**) Control invasomes without terpenoids did not affect bacterial growth. (**B**) Strong growth inhibition of *S. aureus* exposed to invasomes comprising thymol. (**C**) Invasomes encapsulating menthol did not affect growth of *S. aureus.* (**D**–**E**) Growth inhibition of *S. aureus* exposed to invasomes comprising camphor and cineol. (**F**) Statistical evaluation of the measured zones of inhibition validated the significant inhibition of *S. aureus* growth by thymol-containing invasomes compared to all other terpenoid-invasomes and control. * *p* < 0.05 was considered significant (Mann–Whitney test, one-tailed). (n.s. means not significant)

**Figure 8 biomedicines-08-00105-f008:**
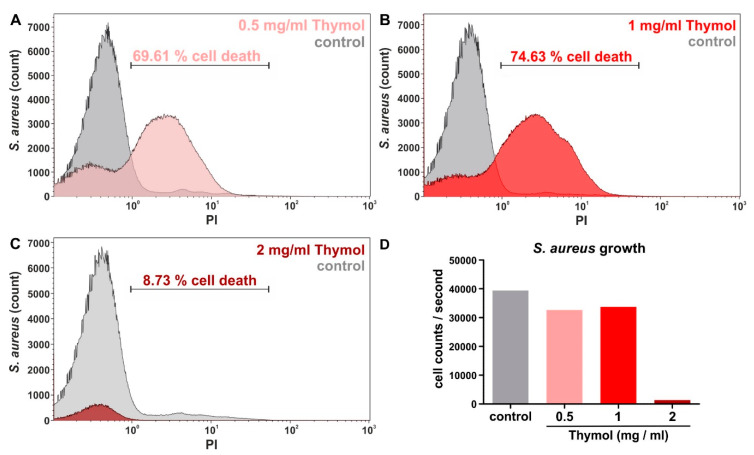
Flow cytometric analysis of cell death validate invasomes encapsulating thymol to be strongly bactericidal against Gram-positive bacteria like *S. aureus*. (**A**,**B**) Compared to untreated negative control, 0.5 mg/mL or 1 mg/mL thymol packaged in invasomes resulted in a profound cell death depicted by PI-staining. (**C**) 2 mg/mL thymol packaged in invasomes resulted in only 9% PI-stained *S. aureus*. (**D**) Assessment of cell flow revealed only around 1000 cells/second in the *S. aureus* population treated with 2 mg/mL thymol-comprising invasomes, suggesting a nearly complete cell death prior to following flow cytometric analysis. PI: Propidium iodide.

**Figure 9 biomedicines-08-00105-f009:**
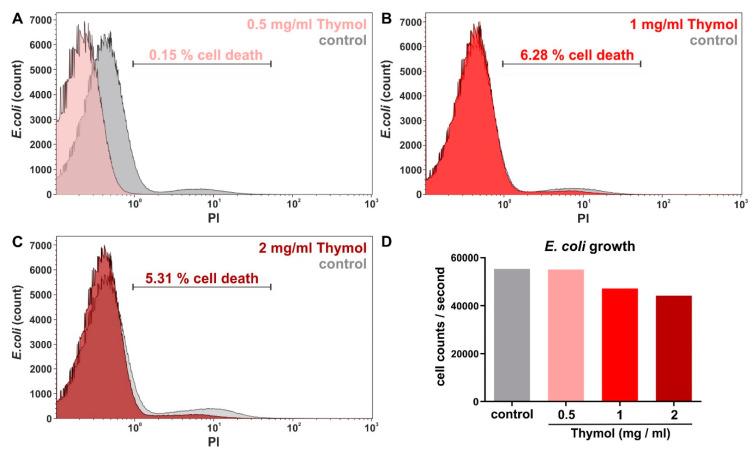
Invasomes encapsulating thymol are not bactericidal against Gram-negative bacteria such as *E. coli*. (**A**–**C**) Compared to untreated negative control, treatment of *E. coli*. with 0.5 mg/mL, 1 mg/mL or 2 mg/mL thymol packaged in invasomes did not result in elevated amounts of cell death. (**D**) Assessment of cell flow revealed no relevant effects of the different concentrations of invasome-encapsulated thymol on the growth of *E. coli*.

**Figure 10 biomedicines-08-00105-f010:**
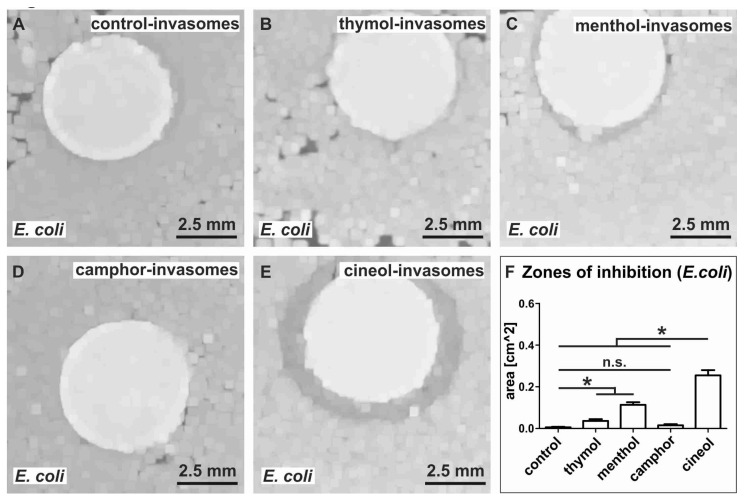
Invasomes encapsulating cineol reveal bactericidal activity against *E. coli* in a growth-inhibition zone assay. (**A**,**B**) Control invasomes without terpenoids or thymol invasomes did not affect bacterial growth. (**C**,**D**) While menthol-invasomes led to a slight inhibition of *E. coli* growth, camphor-invasomes did not affect growth of *S. aureus.* (**E**) Strong growth inhibition of *E. coli* exposed to invasomes comprising cineol. (**F**) Statistical evaluation of the measured zones of inhibition validated the significant inhibition of *E. coli* growth by cineol-containing invasomes compared to all other terpenoid-invasomes and control. * *p* < 0.05 was considered significant (Mann–Whitney test, one-tailed). (n.s. means not significant)

**Figure 11 biomedicines-08-00105-f011:**
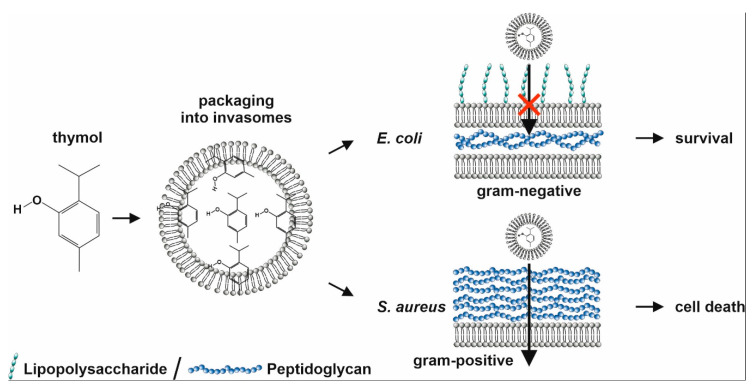
Schematic view on invasomal packaging of thymol and its selective bactericidal activity against Gram-positive *S. aureus*.

**Table 1 biomedicines-08-00105-t001:** Invasome Particle Size and the Polydispersity Index.

Formulation	Particle Size (nm)	Polydispersity Index
Thymol	66 ± 36	0.3 ± 0.05
Camphor	60 ± 32	0.3 ± 0.04
Cineol	86 ± 34	0.2 ± 0.03
Menthol	118 ± 44	0.2 ± 0.03
Control	80 ± 30	0.1 ± 0.02

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
