# Peer review of "Preparation of Terpenoid-Invasomes with Selective Activity against S. aureus and Characterization by Cryo Transmission Electron Microscopy"

_biomedicines, 2020, doi:10.3390/biomedicines8050105_

Round 1

Reviewer 1 Report

The present work focuses on the preparation, characterization and antimicrobial properties of invasomes encapsulating thymol, menthol, camphor and cineol. Although the authors refer to the related literature and apply the methods accordingly, the presentation of their results is difficult to follow. I think the manuscript contains all the relevant information, but the presentation and analysis of the results need to be rethought. Compared to their results the Discussion part seems to be short. Furthermore, I suggest a combined Results and discussion part for easier comprehension.

The detailed comparative analysis of antimicrobial properties of pure terpenoids and nanocarriers encapsulating terpenoids is not completed. Only one sentence refers that antimicrobial properties of pure terpenoids show similar tendencies such us terpenoid-invasomes investigated in the present work (page 14 line 9-10). Furthermore, in the discussion several works are referred to compare some selected properties of encapsulated terpenoids but without any systematic comparison of terpenoid-invasomes with other nanocarriers encapsulating terpenoids. In the discussion, the order is reversed to the results section, which may be due to the fact that the authors try to show the effect of the characteristic properties on antibacterial properties.

The section Conclusion is not mandatory in Biomedicines. However, I really miss it what is the most important result of the work and why it is important. The last two sentences describe the potential application of the described system, however, this last paragraph (literature background of the present work) should be transferred to the introduction part of the manuscript.

Abbreviations are not appropriate. The abbreviation of the words is given in parentheses after the first occurrence of the given word. Subsequently, the abbreviated form is used for the given word. E.g.

page 2 line 38 cryo transmission electron microscopy

page 3 line 16 Transmission electron microscopy (TEM)

page 5 line 2 cryo transmission electron microscopy (TEM)

page 5 line 6 Cryo transmission electron microscopy (TEM)

page 5 line 11-12 TEM: Cryo transmission electron microscopy

page 13 line 19 transmission electron microscopy,

Author Response

Dear Sir or Madame,

Please find enclosed our revised manuscript entitled “Preparation and characterization of terpenoid-invasomes with selective activity against S. aureus by Kaltschmidt et al.. Next to the revised version of our manuscript, please find also enclosed a point-to-point response to the helpful concerns raised by the referee. All changes in the manuscript are highlighted in red. From our point of view, we were able to address all of the referees’ comments.

Thank you very much for your interest in our manuscript, we are greatly looking forward to hearing from you.

Yours sincerely

Bernhard Kaltschmidt (on behalf of Prof. Dr. Andreas Hütten)

Point to point response

Ad Reviewer 1:

We thank for this in depth review and have revised as suggested:

R1:  Furthermore, I suggest a combined Results and discussion part for easier comprehension. A: The authors thank for the kind suggestions, which we are happy to follow in the revised version.

Page 5, line 3: We now combined results and discussion, see chapter 3.

Now we included a discussion of the results on page 5, line 23- page 6, line 7:

During characterization of the newly produced invasomes via cryo transmission electron microscopy, we observed that encapsulation of thymol, camphor resulted in a significant shift towards unilamellar vesicles. We suggest that terpenoids such as thymol might decrease membrane fluidity and thus lead to more unilamellar liposomes. On the contrary, cineol-invasomes revealed a nearly similar distribution between unilamellar and bilamellar vesicles comparable to control. Likewise, cineol-invasomes showed a similar size to control-invasomes, while encapsulation of thymol and camphor led to significantly smaller invasomes and menthol-comprising invasomes were significantly enlarged compared control. We suggest this observation to be based on the elevated water solubility of cineol (3500 mg / L) compared to camphor (1600 mg/ L), thymol (900 mg / L) and menthol (420 mg / L).

Furthermore on page 6, line 20 to page 7, line 17 the new discussion part was included:

In terms of invasomal stability, high Zeta potentials of at least ± 20 mV are generally considered as an indicator for electrostatical and steric stabilization of invasomes [30,31]. Although all produced terpenoid-invasomes showed neutral Zeta potentials in the present study, we observed the presence of stable invasomes using cryo TEM after extrusion. A limitation of our study was that the low values of zeta potential could only be measured with low precision e.g. 3 mV ± 6 for thymol containing invasomes. Furthermore measurements of zeta potentials in nanoemulsions such as invasomes are hampered by measuring limitations arising in diluted samples. Hence plenty of parameters which influence zeta potentials such as viscosity, pH, dielectric constant etc. are not correctly reflected in diluted samples. CryoTEM has to the best of our knowledge not these limitations, since samples with much higher concentrations of invasomes could be analysed in their native diluent [32].

In accordance to our findings, Sebaaly and colleagues reported neutral Zeta potentials of of -3.9 ± 1.9 mV for eugenol-loaded Lipoid S100-liposomes prepared by ethanol injection method. Although the authors demonstrated an increase in liposome size and size distribution after storage in aqueous suspension at 4 °C for 2 months, encapsulation efficiencies of eugenol (86.6 %) were unchanged [33]. We likewise suggest that encapsulation efficiencies of our produced invasomes may not be affected over time, despite their neutral Zeta potentials.

Likewise to providing an increased bioavailability and a more controlled drug release, our approach may also facilitate topical administration of thymol-invasomes due to the high permeability rate of invasomes through the skin [2,21,24]. 

P15, line 15-16: Potential antibacterial mechanisms of invasome formulations with terpenoids are depicted in Fig. 11.

Near the end of results and discussion we added a new part on the bactericidal effects of terpenoids.

P16. Line 4- p17 line 23:

Although terpenoids like limonene, cineole or beta-citronellene have been widely used for formulation of invasomes [26,27], they were mostly applied as permeation enhancer for transdermal delivery and bioavailability and not as bioactive components [2]. In the present study, we focused on the production of invasomes with terpenoids as the bioactive components to inhibit bacterial cell growth. We found our terpenoids-invasomes to be bactericidal against gram-positive S. aureus, with increasing efficiency from cineol- and campher-invasomes (moderate bactericidal activity) to thymol- invasomes showing the strongest bactericidal effects. These findings are in line with the commonly reported bactericidal activity of thymol, campher and cineol [10,12,13]. Interestingly, Mulyaningsih and colleagues reported that exposure of MRSAeven to high concentrations of cineol does not inhibit multi-resistant S. aureus. However, a combination of the terpene aromadendrene with cineol resulted in reduced bacterial cell growth [34]. Extending these findings, we show that encapsulation of cineol into invasomes alone is sufficient for inhibiting growth of S. aureus without the application of additional terpenes. In accordance to the strong bactericidal effects of thymol-invasomes observed here, encapsulation of thymol into other nanocarriers like ethylcellulose/methylcellulose nanospheres was also described to preserve its anti-bacterial activity against S. aureus [35]. In contrast to invasomes encapsulating thymol, camphor and cineol, we observed no anti-bacterial effects for menthol-invasomes against S. aureus, which is contrary to the already described bactericidal activity of menthol [11,12]. With regard to the very low water-solubility of menthol, we suggest the invasomal packaging of menthol to be challenging, in turn affecting its bactericidal activity. In this line, we observed an increased average size of menthol-invasomes compared to all other terpenoid-invasome preparations and control-invasomes lacking terpenoids. In addition polydispersity index as a measurement of the uniformity of invasome size distribution, with a higher value resulting in a broader distribution, was highest with thymol (0.3) and camphor (0.3), suggesting a correlation to antibacterial activity. 

Next to gram-positive bacterial species like S. aureus, we also investigated potential anti-bacterial properties of our invasomes and particularly thymol-invasomes on gram-negative E.coli. Here, cineol-comprising invasomes led to a strong inhibition of E. coli growth, which is in line with our previous observations [13]. In contrast to the strong bactericidal effects against S. aureus, cell growth of gram-negative E.coli was not affected by thymol-invasomes. These observations are contrary to the findings by Salvia-Trujillo and colleagues reporting a bactericidal activity of essential oils of thyme (containing thymol) after incorporation into nano-emulsions [35]. In particular, nano-emulsions comprising essential oils of thyme with heterogeneous droplets sizes between 10 nm to 500 nm were shown to reduce growth of E. coli [36]. However, the authors applied unfractionated essential oils, suggesting a synergistic action of many terpenes to be necessary for bactericidal activity against gram-negative species. Interestingly, Trombetta and coworkers demonstrated that S. aureus appears to be far more sensitive to thymol than E. coli [37], which is in line with our present data. The authors reported a minimal inhibiting concentration of 5.00 mg/ml thymol for E. coli [37], suggesting the concentration of up to 2 mg/ml thymol in the here applied invasomes to be not sufficient for inhibition of E. coli growth. Furthermore, S. aureus is known to secrete pore-forming toxins (PFTs), which were shown to mediate the release of encapsulated clove oil from liposomes. In particular, Cui and coworkers reported that PFTs form pores within the liposome membranes, allowing release of the encapsulated clove oil and facilitating its antibacterial activity. On the contrary, liposomal packaged clove oil had no bactericidal effects on E. coli, which does not secrete PFTs and thus prevents the release of antibacterial essential oil from the liposome [38]. Our present observations may suggest a similar mechanism for thymol-invasomes leading to its selective activity against S. aureus (Fig. 11). In addition, electrophoretic mobility measurement revealed a less soft surface of E. coli compared to S. aureus [39].  The softer surface of S. aureus  mainly comprising peptidoglycan may facilitate entry of thymol-invasomes into the bacterial cells more easily compared to gram-negative E. coli (Fig. 11). Accordingly, we achieved a highly efficient killing of S. aureus with only 0.5 mg/ml thymol encapsulated in invasomes in the present study (Fig. 11).

R1: The detailed comparative analysis of antimicrobial properties of pure terpenoids and nanocarriers encapsulating terpenoids is not completed. Only one sentence refers that antimicrobial properties of pure terpenoids show similar tendencies such us terpenoid-invasomes investigated in the present work (page 14 line 9-10).

A: We are sorry for the misunderstanding but pure terpenoids have in our assays no anti-bacterial activity as we described on page 15,  line:10-13.

We next determined the potential bactericidal activity of non-extruded terpenoids as a control to our encapsulation approach. In contrast to terpenoids encapsulated in invasomes (Fig. 7-10), application of non-extruded terpenoids did not result in growth inhibition of E. coli or S. aureus (data not shown).

R1: Furthermore, in the discussion several works are referred to compare some selected properties of encapsulated terpenoids but without any systematic comparison of terpenoid-invasomes with other nanocarriers encapsulating terpenoids.

A: Thank for this important suggestion, we now addressed this issue more thoroughly  in P17 line 37 till page 18 line 6: There are several nanocarrier systems, encapsulating terpenoids, which are systematically reviewed in (2). The formulation systems encapsulating terpenoids include polymer-based systems such as nano-capsules, nano-particles, nano-fibers and nano-gels. Furthermore, lipid-based systems are frequently used (67% of the formulations), presumably due too the low toxicity. A subgroup of lipid systems are the vesicular systems, which include invasomes. The most investigated biological activity of terpenoids in nano carrier systems is the anti-inflammatory action. Invasomes were used as anti-Acne treatments, hypertension treatment and photosensation therapy (2).

Whereas antimicrobial activity was reported with nano capsules with essential oils from lemon grass, nano emulsions with tea tree oil and penetration enhancing vehicles with essential oil from Santolina insularis. Here we report a novel antimicrobial application of invasome formulations with terpenoids.

R1: In the discussion, the order is reversed to the results section, which may be due to the fact that the authors try to show the effect of the characteristic properties on antibacterial properties.

A: : We now combined results and discussion  and adopted discussion to the order of results.

R1: The section Conclusion is not mandatory in Biomedicines. However, I really miss it what is the most important result of the work and why it is important. The last two sentences describe the potential application of the described system, however, this last paragraph (literature background of the present work) should be transferred to the introduction part of the manuscript.

A: The literature background for our work was now transferred to the introduction, as suggested by the referee.

Page 2, Line 31-37:

In addition to increasing stability of encapsulated compounds, terpenoid-encapsulation systems are widely accepted to be non-cytotoxic and enhance the antioxidant and anti-inflammatory activities of terpenoids ([18,21-23] reviewed in [2] and [24]). For instance, Manconi and colleauques reported the liposomal formulation of thymus essential oil to be highly biocompatible and to counteract oxidative stress in keratinocytes [22]. Thymol encapsulated in nanostructured lipid carriers further showed anti-inflammatory activity in different mouse models of skin inflammation in vivo [21].

We added a conclusion (Page 18 line 1-18):

We conclude that our findings might provide a promising approach to increase the bioavailability of terpenoid-based drugs and might be applicable for treating severe bacterial infections like methicillin-resistant S. aureus (MRSA) in the future. In this regard, the major treatment aims of our formulations include a broad spectrum of applications, ranging from mucosal infections in airway diseases to systemic infections such as sepsis. In this direction we have previously shown that patients with chronic rhinosinusitis have increased levels of S. aureus containing biofilms in the nose [13]. Growth of S. aureus biofilms on the nasal mucosa could be inhibited by 1,8 cineol. Here we extend these findings to thymol containing invasomes, which are much better in its antibacterial activity than formulations with 1,8 cineol (see Fig. 7). Taken together an invasome formulation as described here, containing thymol might be useful as an aerosol spray for pre-operative nose cleaning and might have less side effects in comparison to disinfectants directly applied on the mucosa. As a general use it might be envisaged that invasomes containing thymol or other terpenoids could be used to treat infected surfaces as in nose, lung and skin wounds. Invasomes containing terpenoids might be used in addition or as an alternative to antibiotics.

R1: Abbreviations are not appropriate. The abbreviation of the words is given in parentheses after the first occurrence of the given word. Subsequently, the abbreviated form is used for the given word. E.g.

page 2 line 38 cryo transmission electron microscopy

A: Now cryo transmission electron microscopy (Cryo TEM) (Page 2 line 44).

page 3 line 16 Transmission electron microscopy (TEM)

A: Cryo TEM (Page 3, line 17); Cryo Transmission electron microscopy (Cryo TEM) (Page 3, line 16)

Page 3 line 30 Cryo TEM

page 5 line 2 cryo transmission electron microscopy (TEM)

Cryo TEM image were measured in 3-4 representative Cryo TEM images. For evaluation of bilayer thickness, up to 30 invasomes were measured in 3-4 representative Cryo TEM (Page 4,line 31-32).

page 5 line 6 Cryo transmission electron microscopy (TEM)

Cryo TEM (Page 5, line 21).

page 5 line 11-12 TEM: Cryo transmission electron microscopy

(Cryo TEM) (Page 5, line 8).

page 13 line 19 transmission electron microscopy,

not applicable any more

Reviewer 2 Report

In the manuscript entitled “Preparation and characterization of terpenoid-invasomes with selective activity against S. Aureus” by Kaltschmidt et al. (Biomedicine 779248) the preparation and characterization of invasomes containing various terpenoids were carried out and their antibacterial activity was investigated. In my opinion the manuscript could be accepted for publication in Biomedicines after some modifications. The specific points are:
1.Line 30, page 4: typing mistake und.
2. I am surprised to see the zeta potential values with three significant figures. Nothing about the precision of these measurements is said in the text. How many repetitions were done? The authors should include this information.
3. The same can be said about the percentage of the different invasome populations, invasome sizes and liposomes bilayer thickness. Four significant figures seem too many. The authors should revised the number of significant figures of all the magnitudes given in the manuscript. The errors should be written with only one significant figure, except if the figure is 1, in which case, two significant figures can be used.
4. p should be defined in the text.
5. Nothing is said about the possible influence of the polidispersity of the invasomes on their antibacterial activity.

Author Response

Reviewer 2:

We thank the reviewer for this in depth review and are happy to revise the text as suggested.

R2: 1.Line 30, page 4: typing mistake und. A:This  typo was changed to “and” on page 4 line 30.

R2: 2. I am surprised to see the zeta potential values with three significant figures. Nothing about the precision of these measurements is said in the text. How many repetitions were done? The authors should include this information. A: Thank you for this hint, we have rounded the values of the zeta potential to one significant figure. A limitation of our study was that the low values of zeta potential could only be measured with low precision e.g. 3 mV ± 6 for thymol containing invasomes. Furthermore measurements of zeta potentials in nanoemulsions such invasomes are hampered by measuring limitations  arising in dilute samples. Hence plenty of parameters which influence zeta potentials such as viscosity, pH, dielectric constant etc. are not correctly reflected in dilute samples. CryoTEM has to the best of our knowledge not these limitations, since samples with much higher concentrations of invasomes could be analysed in their native diluent.

Page 3, line 27:  Measurements were repeated ten times.

R2: 3. The same can be said about the percentage of the different invasome populations, invasome sizes and liposomes bilayer thickness. Four significant figures seem too many. The authors should revised the number of significant figures of all the magnitudes given in the manuscript. The errors should be written with only one significant figure, except if the figure is 1, in which case, two significant figures can be used.

A: We thank the reviewer for the advice and made the following changes: We rounded all values to 1 or 2 significant figures as necessary, in the results section (marked in red).

R2: 4. p should be defined in the text.

A: Thank you for this suggestion. We have now included the following text in Materials and Methods, Statistical analysis (p. 4, line 29): The p value is a probability, with a value ranging from zero to one. The first step is to state the null hypothesis, here that the terpenoids do not affect the size of the invasomes and all differences in size are due to random sampling. The p-value is the probability of obtaining results as extreme as the observed results of a statistical hypothesis test, assuming that the null hypothesis is correct. The p-value is used as an alternative to rejection points to provide the smallest level of significance at which the null hypothesis would be rejected. A smaller p-value means that there is stronger evidence in favour of the alternative hypothesis.

  1. Nothing is said about the possible influence of the polidispersity of the invasomes on their antibacterial activity.

A: Thank you for mentioning polydispersity index, which we have now calculated and correlated to antibacterial activity:

In Table 1 the particle size of different terpenoid formulations is depicted as measured in Fig. 5.  When the formulations are sorted from the highest antibacterial activity (thymol) to the lowest (cineol), as measured in Fig. 7, it becomes evident, that the terpenoids with the highest antibacterial activity have the highest polydispersity index also (Table 1).

Table 1: Ivasome Particle Size and the Polydispersity Index

Formulation

Particle Size (nm)

Polydispersity Index

Thymol

66  ±  36

0.3  ±  0.05

Camphor

60 ±  32

0.3  ±  0.04

Cineol

86  ±  34

0.2  ±  0.03

Menthol

118  ±  44

0.2  ±  0.03

Control

80  ±  30

0.1  ±  0.02

For the discussion we included:  In addition polydispersity index as a measurement of the uniformity of invasome size distribution, with a higher value resulting in a broader distribution, was highest with thymol (0.3) and camphor (0.3), suggesting a correlation to antibacterial activity. 

Round 2

Reviewer 1 Report

The authors made effort to improve the quality of the manuscript and answered several questions, therefore, I suggest publishing the work in Biomedicines.